# Nicotinamide Riboside Augments Human Macrophage Migration via SIRT3-Mediated Prostaglandin E2 Signaling

**DOI:** 10.3390/cells13050455

**Published:** 2024-03-05

**Authors:** Jing Wu, Maximilian Bley, Russell S. Steans, Allison M. Meadows, Rebecca D. Huffstutler, Rong Tian, Julian L. Griffin, Michael N. Sack

**Affiliations:** 1Laboratory of Mitochondrial Biology and Metabolism, National Heart, Lung, and Blood Institute, National Institutes of Health, Bldg. 10-CRC, Room 5-3342, 10 Center Drive, Bethesda, MD 20892, USA; jing.wu@nih.gov (J.W.); maximilian.bley@med.uni-heidelberg.de (M.B.); russell.steans@nih.gov (R.S.S.);; 2Department of Biochemistry, Cambridge University, Cambridge CB2 1QW, UK; 3Cardiovascular Branch, National Heart, Lung, and Blood Institute, National Institutes of Health, Bethesda, MD 20892, USA; 4Mitochondria and Metabolism Center, Department of Anesthesiology and Pain Medicine, University of Washington School of Medicine, Seattle, WA 98195, USA; 5The Rowett Institute, School of Medicine, Medical Sciences and Nutrition, Foresterhill Campus, Aberdeen AB25 2ZD, UK

**Keywords:** chemotaxis, macrophage migration, NAD^+^ boosting, nicotinamide riboside, prostaglandin E2, SIRT3

## Abstract

NAD^+^ boosting via nicotinamide riboside (NR) confers anti-inflammatory effects. However, its underlying mechanisms and therapeutic potential remain incompletely defined. Here, we showed that NR increased the expression of CC-chemokine receptor 7 (CCR7) in human M1 macrophages by flow cytometric analysis of cell surface receptors. Consequently, chemokine ligand 19 (CCL19, ligand for CCR7)-induced macrophage migration was enhanced following NR administration. Metabolomics analysis revealed that prostaglandin E2 (PGE2) was increased by NR in human monocytes and in human serum following in vivo NR supplementation. Furthermore, NR-mediated upregulation of macrophage migration through CCL19/CCR7 was dependent on PGE2 synthesis. We also demonstrated that NR upregulated PGE2 synthesis through SIRT3-dependent post-transcriptional regulation of cyclooxygenase 2 (COX-2). The NR/SIRT3/migration axis was further validated using the scratch-test model where NR and SIRT3 promoted more robust migration across a uniformly disrupted macrophage monolayer. Thus, NR-mediated metabolic regulation of macrophage migration and wound healing may have therapeutic potential for the topical management of chronic wound healing.

## 1. Introduction

Caloric restriction dietary interventions confer immunomodulatory effects, including amelioration of inflammatory-linked diseases [1,2,3,4,5,6], blunting of the inflammasome [7,8], and reductions in circulating cytokine and acute-phase reactant levels [9,10,11]. These interventions are not immunosuppressive, with evidence showing that fasting concomitantly improves chronic inflammatory disease without compromising responsiveness to acute infections or wound healing [12]. Furthermore, a fasting mimetic diet increases cytotoxic CD8^+^ tumor-infiltrating lymphocytes that augment chemotherapeutic-induced tumor killing [13].

As these restrictive dietary interventions initiate a myriad of metabolic and signaling effects, more reductionist approaches are required to dissect the components of this biology [14]. As fasting and caloric restriction increase the NAD^+^/NADH ratio, NAD^+^ biosynthesis precursors including nicotinamide, nicotinamide riboside, and nicotinamide mononucleotide are being studied as a component of this biology to uncover mechanisms of action for disease amelioration [15,16]. At the same time, the role of NAD^+^ biology in innate immune cell function is increasingly being recognized as a pivotal intermediate governing immunometabolism and inflammatory function [17,18,19,20]. Macrophages themselves possess a functional NAD^+^ biosynthesis program and a reduction in de novo synthesis of NAD^+^ in macrophages impairs both phagocytosis and the resolution of inflammation. Conversely, restoration of NAD^+^ in these cells augments macrophage reparative function [21]. In parallel, the activation of murine macrophages by LPS requires an intact NAD^+^ salvage pathway to sustain aerobic glycolysis and canonical IL-1β-, TNF-, and IL-6-orchestrated inflammation [22]. In contrast, in human cells, NR blunts the NLRP3 inflammasome [23] and type I interferon signaling [19]. In addition, nicotinamide supplementation reduces inflammatory cytokines in macrophages during differentiation and polarization into reparative macrophages [24]. Together, these data suggest that NAD^+^ supplementation may have distinct monocyte/macrophage immunomodulatory context-specific effects and this redox-sensing metabolic intermediate may play a role in the immunomodulatory effects of reduced caloric loads. In this study, we explored whether and how nicotinamide riboside may modulate macrophage chemotaxis in primary human circulating monocyte-derived macrophages.

Chemotaxis is an integral feature of circulating monocytes to enable their extravasation through the endothelium and for the subsequent differentiation and recruitment of inflammatory or reparative macrophages to sites of infection and injury [25]. This process is governed by the release of chemokines from professional and non-professional immune cells and their subsequent binding to monocytic/macrophage cognate chemokine receptors to initiate intracellular signaling to guide migration [26]. The chemokine ligands are divided into four groups based on the positioning of their first cysteine residues. Chemokine receptors are G-protein-coupled receptors (GPCRs), each with distinct binding affinities for specific chemokine ligands [27]. The migration of human monocytes through the endothelial barrier is dependent on the expression of the chemokine receptors CX3CR1 and CCR2 in response to the endothelial cell ligands CX3CL1 and CCL2 [28]. Additionally, CCR1 and CCR5 may play a role in the inflammation-triggered migration of CCR2^+^ monocytes [29]. After migration and recognition of danger, monocytes differentiate and the pattern of chemokine receptors changes in response to distinctive chemokine signals emanating from the site of injury, inflammation, or infection [30]. Interestingly, macrophages, in response to distinctive chemokines and chemokine receptor signaling, can emigrate to lymph nodes as a component of the reparative process, as seen in the regression of atherosclerosis [31].

In this study, we exposed differentiating human monocytes to nicotinamide riboside and first explored whether their polarization into an inflammatory (M1) or reparative (M2) phenotype was altered. It appeared that monocyte-derived macrophage (MDM) differentiation was similar in the presence or absence of NR as evidenced by the expression of canonical cell surface receptors. However, the levels of CD64, the high-affinity receptor for IgG [32], were significantly reduced, and CD197, a chemokine receptor designated as CCR7, was induced on M1 macrophages in response to NR. To investigate the effects of NR on chemotaxis, we then examined the expression of additional chemokine receptors on M1 macrophages. In contrast to CD197/CCR7, the canonical chemokine receptor repertoire, i.e., CCR1, and CCR5, was not differentially expressed in response to NR. Interestingly, CD197/CCR7, known for its involvement in the emigration of macrophages away from inflammatory sites [31], was induced by NR, suggesting a potential role of NR in the abatement of inflammatory foci.

We pursued this concept in this study and found that NR indeed promoted wound healing and macrophage migration. This was achieved through prostaglandin E2 synthesis and signaling as well as the function of the NAD^+^-dependent sirtuin, SIRT3.

## 2. Materials and Methods

### 2.1. Study Design and Human Subjects in the Clinical Study

An initial clinical study was performed at the National Institutes of Health (NIH) Clinical Center on 35 subjects with an average age of 24 and average body mass index (BMI) of 24. The details of the study have been previously reported and the study is registered at clinicaltrials.gov (NCT02812238) [19].

### 2.2. Human Monocyte Cultures

Primary peripheral blood mononuclear cells (PBMCs) were isolated from human blood by density centrifugation using Lymphocyte Separation Medium (MP Biomedicals, Santa Ana, CA, USA). Human monocytes were negatively selected from PBMCs using the Monocyte Isolation Kit (Miltenyi Biotec, Santa Ana, CA, USA). The monocytes were then plated 1.5 × 10^6^/well onto a 12-well plate (for RNA isolation, Western blotting, or PGE2 measurement) in RPMI media supplemented with 10% human serum (Sigma, St. Louis, MO, USA). Both human blood and elutriated monocytes were obtained from the blood bank of NIH.

### 2.3. Human Monocyte-Derived Macrophage Differentiation and Polarization

Human monocytes were plated and cultured for 5–6 days for macrophage differentiation (M0). On days 6–7, IFN-γ (20 ng/mL) and LPS (1 ng/mL) were added to generate classically activated M1 macrophages. M2 macrophages were achieved by adding IL-4 (20 ng/mL). Forty-eight hours after polarization, M1 or M2 macrophages were used for downstream assays. NR (0.5 mM) and/or PGE2 synthesis inhibitors were added during the 48 h polarization.

### 2.4. Flow Cytometry

Human monocyte-derived macrophages were labeled with antibodies against different surface markers in FACS buffer (PBS with 0.25 mM EDTA and 0.1% BSA) followed by staining with LIVE/DEAD Fixable Yellow stain (Invitrogen, Waltham, MA, USA). The antibodies were purchased from BioLegend (San Diego, CA, USA), BD Biosciences (Franklin Lakes, NJ, USA) and are listed in Appendix A. Data were acquired using FACSymphony (BD Biosciences, Franklin Lakes, NJ, USA), and post-acquisition analysis was performed using Flowjo 9.9.6 (Treestar Inc., Ashland, OR, USA). Analysis excluded debris and doublets using light-scatter measurements and dead cells identified by live/dead stain. The expression of surface markers was determined within the gated population. Statistical significance between two groups was determined using a two-tailed Student’s *t*-test.

### 2.5. Boyden Chamber Migration Assay

Migration of polarized M1 macrophages was analyzed with a modified Boyden chamber (transwell) assay using a Millicell-24 cell culture insert plate with 5 µm or 8 µm pore size (Millipore, Burlington, MA, USA). An equal number of cells were placed in serum-free (0.5% BSA) cell culture media in the upper well while the culture medium of the lower compartment was supplemented with CCL19 (300 nM) (R&D Systems, Minneapolis, MN, USA). After incubation for 6–8 h at 37 °C under 5% CO_2_, transmigrated cells in the lower compartment were collected for cell counts. The number of migrated cells were counted and normalized to the total cell number in the upper well via the CyQuant Cell Proliferation Assay (Invitrogen, Waltham, MA, USA).

### 2.6. Inhibitors, siRNA, and Nucleofection of Human Monocytes

NR (ChromaDex, Los Angeles, CA, USA) was used in cell culture at 0.5 mM for 16–120 h before subsequent assays. COX-2 inhibitor Celecoxib (Cayman, Ann Arbor, MI, USA) or mPTGES-1 inhibitor Cay10526 (Cayman) was used in cell culture at 5 µM or 10 µM for 48 h, respectively. ON-TARGETplus siRNA for knocking down gene expression of SIRT3 and non-targeting control siRNA were purchased from Horizon Discovery (St. Louis, MO, USA). siRNA against human SIRT3 and control siRNA were incubated with a mixture of nucleofection solution (Human Monocyte Nucleofector Kit)(Lonza, Morristown, NJ, USA) and primary human monocytes and placed in nucleofection cuvettes subjected to program Y-010 for the Nucleofector 2b Device (Lonza). RPMI medium (500 µL) was immediately added into cuvettes after nucleofections. Cells were then plated onto a 6-well plate or 10 cm dish and incubated at 37 °C under 5% CO_2_ for 4–5 days for macrophage differentiation followed by M1 macrophage polarization. Plasmid transfection using an empty vector (pcDNA3.1) or SIRT1-, or SIRT3-expression constructs was conducted following the same protocol as siRNA transfection.

### 2.7. RNA Isolation and Quantitative PCR Analysis

Total RNA was extracted from monocytes using the NucleoSpin^®^ RNA Kit (Takara, San Jose, CA, USA) and RNA concentration was measured using the NanoDrop Spectrophotometer (Thermo Fisher Scientific, Waltham, MA, USA). cDNA was synthesized using the SuperScript™ III First-Strand Synthesis System for RT-PCR (Thermo Fisher Scientific) according to the manufacturer’s instructions. Quantitative real-time PCR was performed using FastStart Universal SYBR Green Master (Rox) (Roche, Indianapolis, IN, USA) and run on LightCycler 96 Systems (Roche Holding). Transcript levels of COX-2, mPTGES-1, and 18S rRNA were measured using validated gene-specific primers (QIAGEN, Germantown, MD, USA). Relative gene expression was quantified by normalizing cycle threshold values with 18S rRNA using the 2^−ΔΔCt^ cycle threshold method.

### 2.8. Western Blotting

Human monocytes were lysed using RIPA buffer supplemented with protease inhibitor cocktail (Roche, Indianapolis, IN, USA) and phosphatase inhibitors (Sigma-Aldrich, St. Louis, MO, USA). The lysates were separated by NuPAGE™ 4–12% Bis-Tris Protein Gels (Thermo Fisher Scientific, Waltham, MA, USA) and transferred to nitrocellulose membranes using the Trans-Blot^®^ Turbo™ Transfer System (Bio-Rad Laboratories, Hercules, CA, USA) according to the manufacturer’s instructions. Membranes were blocked with 50% Odyssey Blocking Buffer in PBS-T (0.1% Tween20 in PBS) buffer and incubated with appropriate antibodies overnight at 4 °C. COX-2 antibody was purchased from Santa Cruz Biotechnology (Dallas, TX, USA) or Cell Signaling Technologies (Danvers, MA, USA). Antibodies against mPTGES1 and vinculin were purchased from Millipore Sigma (Burlington, MA, USA). Primary antibody incubations were followed by incubation with IRDye^®^ secondary antibodies for 1 h at room temperature. Immunoblots were visualized and imaged using the Odyssey CLx Imaging System (LI-COR Biosciences, Lincoln, NE, USA). Protein band intensity was measured using Image J software (National Institutes of Health, Bethesda, MD, USA) and normalized to vinculin.

### 2.9. LC/MS Measurement of NAD^+^ and PGE2

Human monocyte cultures were supplemented with 0.5 mM NR or vehicle control for 24 h without LPS stimulation (naïve monocytes) or with 1 ng/mL LPS for 1 h (activated monocytes).

Cells were scraped off the plates and centrifuged at 4 °C for 10 min; cell pellets were then collected and snap frozen in liquid nitrogen. The measurement of NAD^+^ and PGE2 from the cell pellets using LC/MS has been described previously [19]. The polar metabolites in serum samples in the in vivo placebo vs. NR supplementation study were measured using targeted MS/MS as described [20].

### 2.10. PGE2 Measurement from the Cell Cultures

Cell culture supernatants from human monocytes or MDM were collected, centrifuged to remove cells and debris, and stored at −80 °C for later analysis. PGE2 was assayed using the Prostaglandin E2 Parameter Assay Kit (R&D Systems, Minneapolis, MN, USA). Results were normalized to cell number, as determined by the CyQuant Cell Proliferation Assay (Invitrogen, Waltham, MA, USA).

### 2.11. IncuCyte Scratch Wound Cell Migration Live-Cell Analysis

Prior to initiating the scratch wound assay, differentiated M0 macrophages were plated onto a 96-well ImageLock plate at a density of 80 × 10^3^ per well with M1 polarization medium (±0.5 mM NR). Twenty-four hours after plating, IncuCyte^®^ WoundMaker (Sartorius, Bohemia, NY, USA) was applied to create wounds in all wells of a 96-well ImageLock plate by gently removing the cells from the confluent monolayer using an array of 96 pins. After rinsing the wells with medium, culture medium containing CCL19 (300 nM) was added and the plate was placed inside the IncuCyte for live scanning at 2 h intervals for up to 120 h. The data were analyzed for the following integrated metrics: relative wound density (the spatial cell density in the wound area relative to the spatial cell density outside of the wound area at every time point) and wound confluence (a report of the confluence of cells within the wound region).

### 2.12. Statistical Analysis

Statistical analysis was performed using Prism 10 software (GraphPad, La Jolla, CA, USA) and results are presented as mean ± SEM unless otherwise indicated. Comparisons of two groups were calculated using a paired or unpaired two-tailed Student’s *t*-test. Comparisons of more than two groups were calculated using a one-way analysis of variance (ANOVA) with Sidak multiple comparisons test or Dunnett’s multiple comparisons test. For all tests, *p* < 0.05 was considered significant.

## 3. Results

### 3.1. NR Modulates the Expression of M1 Macrophage Markers and Increases CD197/CCR7-Mediated M1 Macrophage Migration

As an initial screening study, we assessed whether NR could modulate the expression levels of canonical M1 and M2 cell surface markers during macrophage polarization. As expected, the M1 population was enriched by a population of CD64^+^CD80^+^ cells compared to naïve (M0) or IL-4 induced M2 macrophages, which showed an increase in a population of CD11b^+^CD209^+^ cells (Appendix A). Interestingly, the polarized M1 cells incubated with NR showed a significant reduction in expression of CD64 and a significant induction of CD197/CCR7 expression (*n* ≥ 11) as determined by FACS analysis (Figure 1A–C). In contrast, the levels of CD80 and M1 chemokine receptors including CCR1, CCR2, and CCR5 were not affected by NR (Figure 1A,C and Appendix A).

As CD197/CCR7 signaling promotes macrophage migration [31], a Boyden chamber was employed as part of an initial screening study to evaluate the functional consequences of NR-mediated upregulation of this receptor. Here, vehicle control and NR-supplemented M1 MDMs were incubated in the upper chamber and the CCR7 ligand CCL19 was introduced into the lower chamber (Figure 2A). In parallel with the NR-mediated induction of CCR7 expression, the rate of migration into the CCL19-enriched chamber was significantly higher for the NR-supplemented macrophages (Figure 2B,C).

### 3.2. NR Increases Levels of the Chemotaxis Mediator PGE2, Which Is Required for NR-Regulated Macrophage Migration

Prior data have shown that prostaglandin E2 (PGE2) has significant immunomodulatory effects [33] and is a major regulator of CCR7-mediated chemotaxis in dendritic cells [34]. We therefore evaluated whether NR modulated the levels of PGE2 in macrophages. Interestingly, NR increased PGE2 levels by ~50% in cultured primary human MDMs (Figure 3A). Furthermore, a separate targeted metabolomics approach, described previously [19], confirmed that ex vivo NR supplementation increased NAD^+^ and PGE2 levels (Figure 3B–D). Additionally, we previously employed NR as an oral supplement in healthy volunteers compared to placebo controls [19,20], and reassessment of these data confirmed that NR significantly increased PGE2 levels in human serum (Figure 3E).

To evaluate whether PGE2 confers this NR-mediated regulation of CCR7 expression, the synthesis of PGE2 was blocked by the cyclooxygenase 2 (COX-2) inhibitor, Celecoxib (CC) or by inhibiting microsomal PGE-synthase 1 (mPTGES-1) with CAY10526 (Cay) (Figure 4A) [35]. Notably, Cay, being a more direct inhibitor of PGE2 synthesis, led to a significant and robust reduction in NR-induced CCR7 expression (Figure 4B) and MDM migration (Figure 4C). However, the broader, more upstream cyclooxygenase inhibitor did not show the same effect (Figure 4B,C). The addition of PGE2 itself induced CCR7 to a similar extent as NR and had an even more robust effect on the migration of M1 MDM (Figure 4B,C). Additionally, NR upregulated the steady-state levels of COX-2 (fragmented forms, which confer enhanced enzymatic activities [36]) without changing the levels of the inducible mPTGES-1 (Figure 4D,E).

### 3.3. Activation of the Sirtuin Deacetylase SIRT3 by NR Drives PGE2 Synthesis and Macrophage Migration

NR, as an NAD^+^ precursor, activates the sirtuin deacetylase enzymes SIRT1 and SIRT3 to modulate immunity [7,37]. To assess whether these enzymes contribute to this regulation, we examined the effect of SIRT1 and SIRT3 overexpression on PGE2 biosynthesis enzymes in LPS-activated primary human monocytes. Interestingly, SIRT3 but not SIRT1 overexpression profoundly increased steady-state levels of COX-2 without affecting levels of mPTGES-1 (Figure 5A,B). Interestingly, the transcript level of COX-2 was not affected by SIRT3 overexpression, suggesting potential post-transcriptional regulation by SIRT3 (Appendix A). Additionally, compared to other sirtuins, SIRT3 overexpression seemed to upregulate the transcript level of mPTGES1, although the protein level was not affected. In parallel, SIRT3 overexpression increased the level of PGE2 in human M1 MDMs (Figure 5C) and promoted migration in response to CCL19 (Figure 5D). Conversely, siRNA knockdown of SIRT3 in human M1 macrophages attenuated NR-mediated migration relative to control siRNA cells (Figure 5E).

The scratch test, as a model of wound healing [38], was then employed to validate the effects of NR and SIRT3 on macrophage migration. Figure 6A compares the degree of wound healing on Day 0 and Day 2 between vehicle control and incubation with NR. In the presence of CCL19, NR increased the relative degree of migration (relative wound healing) and the rate of wound confluence (Figure 6B,C). In parallel, the relative degree of wound density (migration) was blunted by siRNA knockdown of SIRT3 (Figure 6D,E) and enhanced by SIRT3 overexpression (Figure 6F,G).

## 4. Discussion

A substantial body of evidence supports that NR, as an NAD^+^-boosting supplement, has anti-inflammatory effects [19,20]. Nevertheless, its role in regulating myeloid cell migration has not been previously investigated. In this study, we show that in human macrophages, NR induces surface expression of the chemotaxis CD197/CCR7 receptor and levels of its lipid mediator PGE2 via upregulation of cyclooxygenase 2 and functionally increases macrophage migration and wound healing in a SIRT3-dependent manner. These data expand the repertoire of anti-inflammatory effects of NAD^+^ boosting.

The mechanisms of action of NAD^+^ boosting in immunoregulation are extensive, exemplifying the different roles of the NAD^+^/NADH redox system and the direct role of NAD^+^ as a cofactor or substrate for different enzyme systems within leukocytes. These regulatory effects include epigenetic control, immunometabolic signaling, and effects on intracellular organelle homeostasis and signaling as well as direct effects on sirtuin enzyme functions. Hence, although the effects on myeloid cell migration may be multifactorial, in this study, we initially focused on the potential role of sirtuins. Interestingly, the sirtuin enzymes have vastly divergent Michaelis constant (Km) values for NAD^+^ [39]. Furthermore, comparing sub-compartment NAD^+^ levels, the most likely sirtuin enzymes which can hypothetically be modified by NAD^+^ boosting are SIRT1, SIRT3, and SIRT5 [39]. NR, via NAD^+^ boosting, has been shown to activate both SIRT1 and SIRT3 activity [23,40]. Our study finds that SIRT3 rather than SIRT1 augments COX-2 levels. Furthermore, the genetic gain and loss of SIRT3 levels also show that SIRT3 increases PGE2 levels and modulates macrophage migration and wound healing. SIRT1 has previously been shown to promote wound healing; however, in that study, the mechanism appeared to be via an AMPK-dependent pathway [41].

PGE2 is the most abundant eicosanoid in the inflammatory milieu, and its levels reflect the balance between its COX-2-regulated synthesis and 15-hydroxyprostaglinadin dehydrogenase-driven degradation [42]. It has predominantly anti-inflammatory effects on innate immune cells, although it does promote T_H_2, T_H_17, and regulatory T-cell responses [42]. Its different effects are orchestrated through different receptors on different cell types and are regulated in response to a variety of concurrent signals [43]. Given this complexity, the overall effects of the levels of PGE2 will ultimately be best characterized in vivo in the context of infective, inflammatory, or autoimmune conditions. Here, given the chemotaxis effects of PGE2 [34], a more reductionist approach was initially taken to directly explore this eicosanoid intermediate’s effect on macrophage migration. Our data uncover that both in vivo NR supplementation and ex vivo NR increase PGE2 levels. Furthermore, cell culture studies in primary human monocytes support that this is due to the upregulation of COX-2 and that this program is mediated in part by the canonical NAD^+^-boosting responsive sirtuin, SIRT3. At the same time, it is interesting to note that in a different context, the knockdown of SIRT3 promotes inflammation-linked chemotaxis with the concomitant induction of PGF_2α_ [44].

COX-2 is an integral membrane-bound inducible enzyme usually localized to the endoplasmic reticulum or outer nuclear envelope [45]. However, in cancer cells, it localizes to mitochondria where it serves an anti-apoptotic function [46]. Furthermore, multiple studies have explored the regulation of COX-2 at the transcript, signaling, and post-translational degradation levels [47,48]. In this study, we found that NR did not alter the transcript levels of COX-2 but rather that NR and SIRT3 overexpression increased its steady-state protein levels. Furthermore, in this study, we did not determine COX-2 subcellular localization. Subsequent studies will also need to explore either SIRT3-mediated signaling or how its deacetylation effects modify COX-2 post-transcriptional regulatory events.

Recent evidence supports that the solute carrier 29 (SLC29) family members, known as the equilibrative nucleoside transporters (ENT1, 2, and 4), facilitate NR uptake into cells [49]. Interestingly, as delineated in the Human Protein Atlas, human macrophages express SLC29A1, 2, and 4, while monocytes express SLC29A1. Although we did not explore these mechanisms of uptake, we previously employed LC/MS analysis to examine the effects of NR on monocytes and found that the levels of NAD^+^ approximately doubled in response to in vitro NR supplementation [19]. It is important to note that the dosing of in vitro NR supplementation (0.5 mM) is orders of magnitude higher than the nM levels of intracellular NR [50]. However, data show that in blood cells, 0.5 mM in vitro NR mimics the in vivo elevation of NAD^+^ levels achieved with oral NR supplementation in human subjects [19,50].

Chronic diabetes is linked to poor wound healing [51] and NR has been shown to improve insulin resistance in prediabetic subjects [52]. Whether NR or other NAD^+^-boosting agents could have a beneficial effect on wound healing in this disease is an intriguing concept. In this regard, nicotinamide is available as a topical ointment, and pilot studies have shown its benefit against atopic dermatitis and psoriasis [53]. Taken together, these data support that the evaluation of topical NAD^+^-boosting treatments for diabetic wound healing warrants further investigation.

In conclusion, this study identifies a new mechanism whereby NAD^+^ boosting with nicotinamide riboside promotes an anti-inflammatory effect in macrophages by promoting PGE2-mediated cell migration. This effect appears to be mediated in part by the canonical role of NAD^+^ as a cofactor for the activation of SIRT3. Consequently, SIRT3 was shown to promote the synthesis of PGE2 by increasing the steady-state levels of the inducible enzyme cyclooxygenase 2, which facilitates PGE2 synthesis. It is noteworthy that in vivo NR supplementation in humans results in the elevation of circulating levels of PGE2. Further investigation is required to determine whether this effect is specifically limited to synthesis in myeloid cells. As topical NAD^+^-boosting agents are available, their application to wounds in diabetic patients is a feasible approach to assess whether this biology is operational in a significant human disease.

## Figures and Tables

**Figure 1 cells-13-00455-f001:**
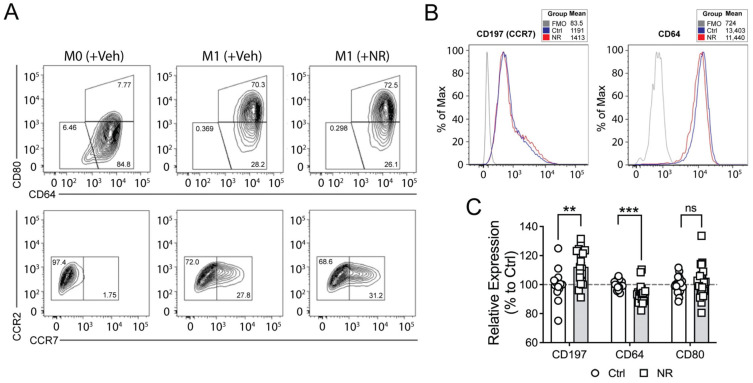
NR administration differentially regulates the expression of M1 macrophage markers in human MDMs. (**A**) Representative flow cytometry contour plots showing CD64/CD80 expression (**top** panel) and CCR2/CCR7 expression (**bottom** panel) in human M0 or M1 MDMs treated with vehicle or NR for 48 h. The gates represent surface marker expression as a percentage of living cells. The population of CCR7^+^ macrophages was increased following NR administration. CCR7 was not detected in M0 macrophages. CCR2 was low in both M0 and M1 macrophages. (**B**) Representative histograms showing cell surface expression of CCR7 (CD197) and CD64. Data shown are representative of 11–12 independent experiments. (**C**) Quantification of relative surface expression of CCR7 (CD197), CD64, and CD80 in human M1 MDM treated with vehicle (Ctrl) or NR. NR increased the surface expression of CCR7 (CD197), decreased CD64, and had no effect on CD80. Data are represented as mean ± SEM. ** *p <* 0.01; *** *p <* 0.001; ns, not significant. Unpaired two-tailed Student’s *t*-test.

**Figure 2 cells-13-00455-f002:**
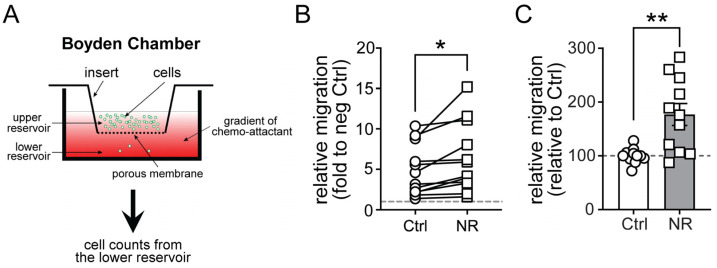
NR administration increases CCR7 (CD197)-mediated migration in cultured human M1 MDMs. (**A**) Diagram showing the migration assay using the Boyden chamber apparatus followed by cell counting from the lower reservoir. (**B**) Pairwise comparison of relative migration in vehicle (Ctrl)- or NR-treated human M1 MDMs in response to CCL19. Each line represents one experiment. Human MDMs were derived from a total of 8 healthy subjects. (**C**) Quantification of relative migration in vehicle (Ctrl)- or NR-treated human M1 MDMs in response to CCL19. NR application increased CCL19/CCR7-mediated migration by ~70%. Data are represented as mean ± SEM. * *p <* 0.05; ** *p <* 0.01. Unpaired two-tailed Student’s *t*-test.

**Figure 3 cells-13-00455-f003:**
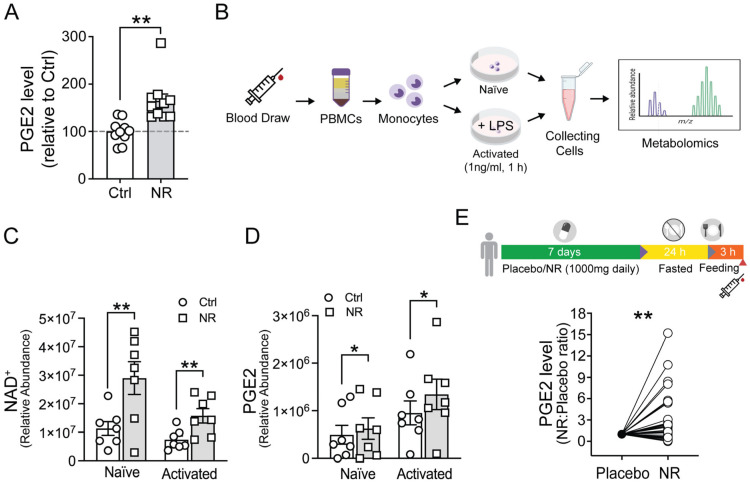
NR administration increases the PGE2 level in cultured human monocytes, MDMs, and human serum. (**A**) The PGE2 level in human primary M1 MDMs treated with vehicle or NR for 48 h (*n* = 10) was measured by the Parameter PGE2 Immunoassay (R&D systems). Human MDMs were derived from 3 healthy subjects. (**B**) Scheme showing unstimulated (naïve) and LPS-stimulated (activated) human monocytes treated with vehicle or NR for 24 h and collected for metabolomic analysis. (**C**,**D**) Relative abundance of NAD^+^ and PGE2 in vehicle control and NR-supplemented groups determined by metabolomics analysis (*n* = 10 healthy subjects). (**E**) (Inset) Design of the clinical protocol. The horizontal bar depicts volunteers consuming NR or placebo for 7 days followed by 24 h of fasting and 3 h of refeeding. The syringe symbol depicts the blood draw time point for serum collection and metabolomics. The PGE2 level in human serum after in vivo NR administration compared to placebo control was measured in a cohort of 36 healthy subjects. Data were analyzed using an unpaired two-tailed Student’s *t*-test (**A**,**C**,**D**) or paired two-tailed *t*-test (**E**). All data are represented as mean ± SEM. * *p* < 0.05; ** *p* < 0.01.

**Figure 4 cells-13-00455-f004:**
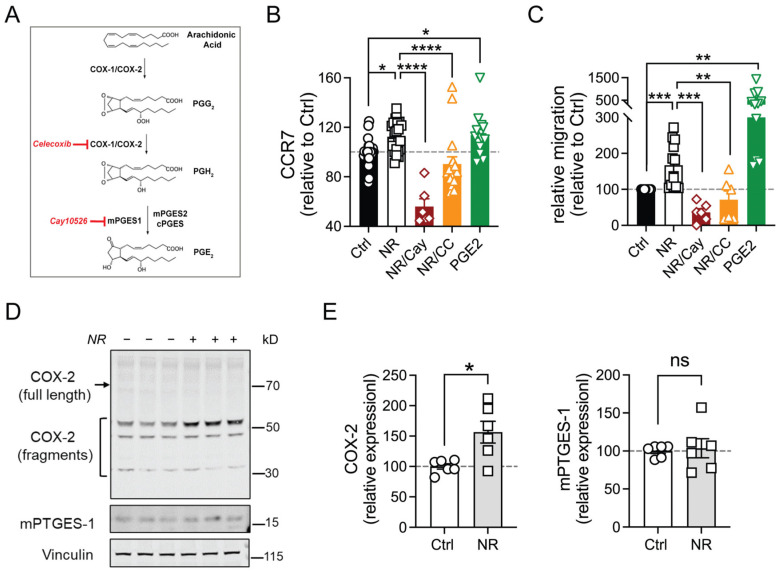
NR-mediated increases in CCR7 expression and CCL19-induced migration are attenuated by PGE2 synthesis blockers. (**A**) Diagram showing the PGE2 synthesis pathway and inhibitors used in this study. (**B**) Relative surface expression of CCR7 was measured by flow cytometry in human M1 MDMs treated with vehicle, NR (±inhibitor: Celecoxib (CC) or Cay10526 (Cay)), or PGE2 during the 48 h polarization. (**C**) Relative migration of human M1 MDMs treated with vehicle, NR (±inhibitor), or PGE2 in response to CCL19. (**D**) Representative immunoblot of PGE2 synthesis enzymes from human M1 MDMs treated with vehicle or NR for 48 h during the polarization. (**E**) Quantification of the COX-2 or mPTGES-1 level in vehicle (Ctrl)- or NR-incubated M1 MDMs, normalized to the vinculin level (*n* = 6 replicates, two experiments). Data were analyzed using a one-way ANOVA followed by Dunnett’s multiple comparisons test (**B**,**C**) or unpaired two-tailed Student’s *t*-test (**E**). All data are represented as mean ± SEM. * *p* < 0.05; ** *p* < 0.01; *** *p* < 0.001; **** *p* < 0.0001; ns, not significant.

**Figure 5 cells-13-00455-f005:**
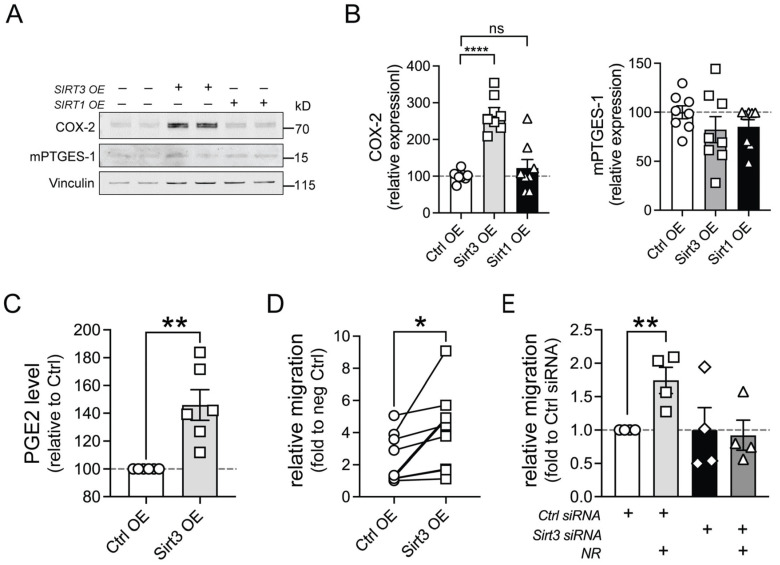
SIRT3 is required for PGE2 synthesis and CCL19-induced migration elicited by NR administration. (**A**) Representative immunoblot of PGE2 synthesis enzymes from LPS-activated human monocytes transfected with an empty vector or SIRT1- or SIRT3-expression plasmids. (**B**) Quantification of the COX-2 or mPTGES-1 level in empty vector (Ctrl OE)-, SIRT1-, or SIRT3-transfected monocytes (*n* = 8 replicates from three experiments). (**C**) PGE2 levels from human M1 MDMs transfected with an empty vector (Ctrl OE) or SIRT3-expression plasmid (SIRT3 OE) measured using the Parameter PGE2 Immunoassay. Human M1 MDMs were derived from 6 healthy subjects. (**D**) Relative migration of human M1 MDMs transfected with an empty vector (Ctrl OE) or SIRT3-expression plasmid (SIRT3 OE) in response to CCL19. Human M1 MDMs were derived from 8 healthy subjects. (**E**) Relative migration of vehicle- or NR-treated human M1 MDMs transfected with either control siRNA or SIRT3 siRNA in response to CCL19. Human M1 MDMs were derived from 4 healthy subjects. Data were analyzed using a one-way ANOVA followed by Dunnett’s multiple comparisons test (**B**,**E**) or an unpaired two-tailed Student’s *t*-test (**C**,**D**). All data are represented as mean ± SEM. * *p* < 0.05; ** *p* < 0.01; **** *p* < 0.0001; ns, not significant.

**Figure 6 cells-13-00455-f006:**
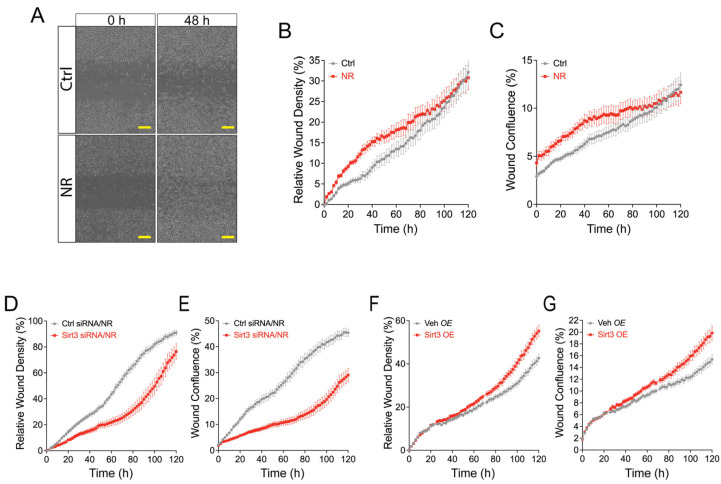
NR facilitates collective cell migration in a SIRT3-dependent manner in human M1 MDMs during wounding. (**A**) Human MDMs were subjected to standardized wounding and wound closure was monitored by an IncuCyte Live-Cell Analysis System. The images show closure of the wound for up to 48 h post-wounding in control vs. NR-treated MDMs. The scale bars denote 200 μM. (**B**,**C**) Relative wound density (RWD) and wound confluence (WC) showing wound closure of human MDMs treated with vehicle (control) or NR over time. (**D**,**E**) RWD and WC showing temporal wound closure of NR-treated human MDMs transfected with control siRNA or SIRT3 siRNA. (**F**,**G**) RWD and WC showing temporal wound closure of human MDMs transfected with empty vector (Veh OE) or SIRT3 (SIRT3 OE).

## Data Availability

The full unedited Western blots in Figure 4 and Figure 5 and Excel spreadsheets for the time-lapse data in Figure 6 are accessible via Figshare with the dataset identifier: https://doi.org/10.25444/nhlbi.24994724.

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
