# Peer review of "Nicotinamide Riboside Augments Human Macrophage Migration via SIRT3-Mediated Prostaglandin E2 Signaling"

_cells, 2024, doi:10.3390/cells13050455_

Round 1
Reviewer 1 Report
Comments and Suggestions for Authors
Obesity-related inflammation is a problem in contemporary society. In this inflammation, immunosuppressive change usually occurs and it causes several diseases such as cancer. In this manuscript, the authors focused on NR, which is a recently discovered nicotinamide derivative. The authors clearly showed the macrophage migration by NR supplementation via SIRT3.
The experiments were well-demonstrated and the logic seemed to be reasonable. The reviewer points out one thing that is recommended to be included in the discussion.
The point is how cells can use NR. Compounds that have ribose are considered hard to permeate cells. So, cells use specific transporters for the corresponding chemicals. For example, NR has been demonstrated to be imported into HEK293 cells via SLC29 and SLC28 families
Kropotov, A. et al.. Int J Mol Sci 2021, 22, 1391, doi:10.3390/ijms22031391.
The expression patterns of transporters depend on cell types, so the confirmation of the pathway of intake of these nutrients to cells is thought to be a fundamental matter in this field.
If you have some data, it is recommended to show it in the experimental section. If not, it should be included in the discussion section referencing some articles. (If necessary, the knowledge about this matter should be written in the Introduction section, too.)
Minor point
In fig 5A, SIRT3 OE markers “+” was disappeared.
Reviewer 2 Report
Comments and Suggestions for Authors
Nicotinamide riboside (NR) has potential as a therapeutic intervention for various metabolic diseases. In this manuscript the authors investigated the underlying mechanisms of NR in wound healing and macrophage migration. This is an interesting and important study and can improve the understanding of the mechanistic role of NR. I have minor remarks regarding the reporting of data (detailed below).
1. In the introduction, caloric restriction/intermittent fasting can be summarized in few sentences to maintain the flow of the manuscript.
2. Figure 6 A: Scale bars should be denoted on the picture.
Reviewer 3 Report
Comments and Suggestions for Authors
In this study, the authors illustrated that the nicotinamide riboside (NR) enhances human macrophage migration through SIRT3-mediated prostaglandin E2 signaling. They isolated the primary peripheral blood mononuclear cells (PBMCs) from human blood and treated them with 0.5 mM NR. The study is well-designed and conducted, but lacks background information and explanation regarding the chosen concertation.
Here are my comments:
1. Why did the authors choose a concentration of 0.5 mM for the experiments?
2. According to pharmacokinetic studies, the blood concentration of NR from recommended daily intake is in the nM level (https://doi.org/10.1371/journal.pone.0186459). How is it possible for the physiological effects and mechanisms obtained in this study using a concentration over 500 times higher to occur in the human body in vivo?
Round 2
Reviewer 1 Report
Comments and Suggestions for Authors
The authors added discussion properly. The manuscript is worth publishing now.
Reviewer 3 Report
Comments and Suggestions for Authors
The author has responded to my questions very well.